# Chilean Teacher Educators' Conceptions on the Absence of Women and Their History in Teacher Training Programmes. A Collective Case Study

Jesús Marolla Gajardo [1,*] , Jordi Castellví Mata [2] and Rodrigo Mendonça dos Santos [3]

1   Faculty of Education, University of The Americas, Santiago de Chile 8370040, Chile
2   Faculty of Education, International University of La Rioja, 26006 La Rioja, Spain; Jordi.Castellvimata@unir.net
3   Department of Curriculum & Pedagogy, Faculty of Education, University of British Columbia, Vancouver, BC V6T 1Z4, Canada; rodrigo.santos@ubc.ca
*   Correspondence: jmarolla@udla.cl

**Abstract:** Schools must assume a clear position that considers gender perspectives and studies in their programmes' construction as well as in discourses and practices produced and reproduced in the school context. Social sciences education is a key area that enables the creation of tools to reflect and foster social justice practices in face of violence against women. In this article, we focus on some reflections of social sciences education professors in Chile. Specifically, we discuss the limitations they face to include women and women issues in their classes. The methodology utilised is Collective Case Studies. The methodology used has a socio-constructivist approach and critical theory perspective, seeking to understand the structures of meaning around the invisibility of women and their history. Among the results, the willingness of professors to include and transform their practices towards perspectives that promote inclusion and social justice stands out. However, they have different limitations, such as excessive workload, the tradition already present in teacher education programmes, and the rigidity of the hegemonic and patriarchal structures.

**Keywords:** teacher training; women; education; conceptions

## 1. Introduction

From different theoretical perspectives in education, an urgency has been manifested that schools, teachers, and other educative actors rethink their practices and behaviours to subvert traditional gender structures (Levstik and Barton 2005; Crocco 2008; Lerner 1981; Lomas 2002; McIntosh 1983; Gràcia et al. 1993; Woyshner 2002). Schools and other educational institutions normalise traditional parameters regarding gender that perpetuate discrimination, segregation, stereotypes, and prejudices (Foucault 1976), reproduce discursive practices around inequalities, and marginalise women within the concept of diversity (Marolla 2019b; Ortega-Sánchez and Pagès 2018; Ortega-Sánchez et al. 2020). It must be considered that, from an educational and historiographical tradition, there are structures that have been responsible for producing and reproducing inequalities on the basis of gender (Massip et al. 2020; Sant and Pagès 2011).

Therefore, this study aims to reveal the gender structures manifested by professors when planning and executing the teaching and learning practices of history and social sciences. In this way, we expect to understand, know, and analyse spaces of possibility and the obstacles described by professors on the teaching staff in the face of the inclusion of women and their history in their the practices. This study carries practical implications for the fundamental interest and thus is related to delivering paths and ideas to teachers, as well as to management and curricular planning teams. We suggest that they reflect on their own gender constructions and generate transformations in the face of the inequalities that have been perpetuated in society.

It is of great relevance to deliver ideas that address the limitations discussed by the study's participants. Despite the various ideas about innovation and the educational advantages that exist regarding the inclusion of women and their history in the teaching of history and social sciences, there are different barriers that traditionally affect teaching (Sant and Pagès 2011; Marolla 2019b; Ortega-Sánchez et al. 2020). Strictly speaking, they refer to hierarchical gender structures, derived from the patriarchal structures themselves, that have normalised such hierarchies. Within the heterosexual matrix (Foucault 1976), it is men who can express themselves in society and public spaces.

For the study, two main objectives guided the research:

1. Interpret the content of the professors' narratives about traditionality and spaces, in order to identify and generate changes in the teaching of history and social sciences.
2. Analyse the main advantages and limitations defined by participants when trying to include work on the inclusion of women and their history in the training of history and social sciences teachers.

## 2. Theoretical Framework

### 2.1. Gender Structures in Educational Contexts

Social studies, as an educational field of study, is concerned with the issue of gender structures in educational contexts. Works such as Luque and Rodríguez (2015), Marolla (2019a, 2019b), Massip and Castellví (2019), Ortega-Sánchez and Pagès (2018), Sant and Pagès (2011), Hahn et al. (2007), Schmeichel (2011), among others, have concluded that current educational practices must be rethought from current gender issues with the aim of including women and sexuality, and gender dissidence in the teaching and learning processes. These works coincide with the fact that curriculum, textbooks, and teachers' attitudes are conditioned by patriarchal structures that exclude topics such as human diversity, social issues, and the history of women (Fernández Valencia 2004; Valencia 2006; Winslow 2013). It is still a pending task to understand and unveil the structures under which women are included and problematized within history and to work from that approach (Woyshner 2002; Woyshner and Schocker 2015). Except for incipient investigations (Ortega-Sánchez and Pagès 2018; Marolla 2019b, 2019a; Ortega-Sánchez 2017), it is difficult to find works in the educational field that deal with the groups that have been invisible throughout history. Furthermore, as Marolla (2019a) suggests, it is essential that studies focus on teacher practices, student learning, and the processes in classrooms that include or exclude women.

Some studies, such as Sant and Pagès (2011), Pagès and Villalón (2013), Cerecer and Pagès (2014), Pinochet (2015), Marolla (2015), and Ortega-Sánchez (2017), have shown that women and their history have been approached from the margins and absences, that is, many times as appendices or curious facts in a history built from masculinity. History and the teaching of history, the authors agreed, highlight the actions of powerful men in politics, the economy, and war. However, despite the research carried out on this topic, the problem persists, and women and their history continue to be taught from traditional logics and with male protagonists in a heteronomous story (Marolla 2019a, 2019b; Ortega-Sánchez and Pagès 2018; Ortega-Sánchez et al. 2020; Schmeichel 2011).

### 2.2. Including Women and Women's History in Teaching Practices

Women have been made invisible in a myriad of processes, and, when present, they are represented with prejudice and traditional gender stereotypes (Marolla 2019a). Scott (1996) stated that history is written excluding women and gender studies, and it is framed in patriarchal structures. At the same time, masculinity is configured under symbols and practices excluding women's actions. According to Butler (1997, 2011, 2016) and Foucault (1976), the conceptualisation of identities is conditioned to multiple social, political, and cultural intersections that conform them. Bourdieu and Passeron (2018), Foucault (1976) and Wittig (1992) agreed that concepts are understood from perspectives such as class, race, sex, or ethnicity, and others that entail control, segregation, and marginalisation. Gender

is not constructed in a coherent way; on the contrary, it depends on the perspectives previously named that, in general terms, are determined by heterosexual conventions on the ways of thinking and acting (Butler 2011, 2004, 2016).

Butler (1997) considered that it is necessary to reflect on the oppressive productions and reproductions formulated by the patriarchy. One of the aims of this reflection must be the rejection of fixed and immutable conceptions about identity, understanding that such aspects abide by power relations within determined sociohistorical contexts. For that reason, there are several historical movements that have sought to make women's actions and issues as visible as those of men (we refer to the actions transmitted from a tradition where protagonists are reflected in issues of politics, the economy, and war) (Massip et al. 2020). There are efforts to represent women from oppression and marginalisation, creating new periodisation and narratives departing from feminine historicity (Scott 1996). In the educational field, Giroux (2018) stated that, by including women and gender issues in teaching education, we can recover their history and their voices. To rethink education is fundamental to understanding concepts such as reproduction, hegemony, or gender hierarchies. In this way, it is possible to build new discourses and narratives, new political and cultural vocabularies that define different expressions and identities.

Research that inquires on the presence and absence of women in school curriculums and textbooks has shown that women's stories are entirely left out (Luque and Rodríguez 2015; Valencia 2006; Sánchez and Pintado 2006; Crocco 2008; Sant and Pagès 2011; Marolla 2015, 2019a, 2019b; Schmeichel 2011). As Banks (2008) stated, it is necessary to reflect on how knowledge has been constructed and how teacher education and teaching practices are being developed in relation to the inclusion and exclusion of women, their diversity, and their stories. Despite the importance of teacher education programmes for transforming educational practices, there are very few studies that inquire about the inclusion of women in history and social sciences teacher education (Crocco 2015b; Hess 2002; Levstik and Groth 2002; Woyshner 2002). As a consequence, teacher education programmes neither consider women nor their stories and narratives (Gutiérrez 1998; Vavrus 2009). Androcentrism is a historical and current issue that urgently requires our attention (Marolla 2019a).

Teachers are not educated for considering women's perspectives in education, and they think and plan their practices according to the *socially most valued model*, which is a male-centred model (Apple 1990; González 2002). As a consequence, androcentrism, sexual stereotypes, and the universality of male values as social and gender structures are transmitted, particularly in history and social sciences education (Hubbard 2013; Vázquez 2003) and school organisation, dynamics, teaching practices, and other related aspects, reproducing stereotypes and gender prejudices.

As teachers act as agents of transmission of values, they are capable of transmitting stereotypes, prejudices, and gender inequalities. Thus, it is fundamental to educate them to produce changes in education and to not reproduce gender inequalities by being attentive to the approaches and perspectives under which they teach history and social sciences. To generate social and educational change from a gender issues perspective, it is necessary to promote discourses that potentiate inclusion, empathy, diversity, and the construction of spaces that enable the empowerment of people who have been marginalised in the narratives (Sánchez and Pintado 2006; Heimberg 2005).

According to Crocco (2008) and Giroux (2018), gender problems obey patriarchal, androcentric, and heteronormative structures (Butler 2011; Foucault 1976) under which the Western educational system is organised. As a result, many gender issues could be shared between different educational contexts. The current study is positioned within the context of teacher education in Chile, the details of which we explain in the following section.

## 3. Methodology

The research we present is framed in a socio-constructivist approach, understanding reality as "a product of a symbolic dynamic that is generated in many social instances" (Stake 2013, p. 125). Along with Álvarez-Gayou (2003), Bryant and Jary (2014), Bisquerra

and Alzina (2004), and Simons (2009), we understand the construction of knowledge as the attribution of meaning to a reality that has been socio-historically determined through contextual interaction. Thus, this research seeks to understand structures of significance around the absence and presence of women and their history in teacher education programs and teacher practices, characterising, defining, and explaining the construction of social representations and discursive practices in history and social sciences teaching (Ortega-Sánchez 2017).

This research is based on a collective case study (Stake 2013; Simons 2009), focused on exploratory (Yin 2009) and interpretive (Simons 2009) case studies. It takes a qualitative approach and, as such, is intended to explain and give relevance and complexity to the research topic (Álvarez-Gayou 2003; Creswell 2014; Stake 2013; Simons 2009). In order to comprehend the content of teacher educators' discourse, we used the principles of critical theory (Cohen et al. 2013; Álvarez-Gayou 2003), as well as the conceptions of feminist research (Cohen et al. 2013). The research is also framed in the transformational field (Creswell 2014), since this perspective intertwines political aspects, transformation, and the fight against oppression (Bryant and Jary 2014).

The participants that took part in the analysis and the reflection were three female and two male teacher educators in social sciences of different universities in Santiago de Chile. Following Ortega-Sánchez (2017), the participant selection depended on the researcher's criteria and not on a random sample selection, thus the selection criteria used were based on the degree of familiarity of participants to previously established objectives and categories of the analysis (Gómez et al. 1996).The sampling technique used had a non-probabilistic and non-stratified character (Hernández Sampieri et al. 2014), based on cohabitation (Cohen et al. 2013; Simons 2009; Creswell 2014), requiring participants to adequately meet the conditions for the purposes of the research (Table 1). For the selection of such participants, the following criteria were followed:

(a)    Predisposition: Participants show their willingness to share their experiences.
(b)    Type of university where they work (public–private).
(c)    Previous training: all participants are graduates in history and education.

**Table 1.** Participants.

| Fictitious Name | University | Work Experience | Age |
| --- | --- | --- | --- |
| Stella | Public | More than 15 years. | 36 |
| Polly | Private | More than 10 years. | 32 |
| Tammy | Public | More than 10 years. | 34 |
| Vince | Private | More than 15 years. | 35 |
| Fred | Public | More than 15 years. | 35 |

For the presentation and discussion of the results, fictitious names were adopted to preserve participants' anonymity. This was done following the ethical criteria established by the Declaration of Helsinki. The participants signed an informed consent prior to the study, counting on the security of anonymity, and the security and permanent protection of the integrity of each person.

To obtain the information, semi-structured interviews were used, and the Bisquerra and Alzina (2004) method was used around the interviews as conversations. In this method of interviews and conversations, the aim was to create a climate of trust with the aim that informants could share their experiences in a fluid way. Aspects such as empathy and dialogue in the interviews were essential in obtaining valuable data for our research. Initially, some questions were prepared and tested under the validation of experts in social science teaching. Afterward, the questions were shared with the people to be interviewed (Álvarez-Gayou 2003). In addition, we use the "probes" strategy, following the approaches of Cohen et al. (2013), creating space in order to request clarifications or depth in the answers provided, which could lead to extensions, greater details or explanations of what

participants have said in the interviews. The audio of the interviews was recorded with an audio system and later transcribed. The transcripts were sent to the participants, following the ethical criteria, in order to give the interviewees the possibility of adding data or requesting the deletion of information that they considered could affect their integrity.

To carry out the qualitative analysis of the information, the four phases proposed by Creswell (2014), Stake (2013) and Simons (2009), were followed: (1) reduction of the data, where a rigorous selection of the data was made in order to facilitate the information for analysis facilities; (2) analysis of the data and their organisation into categories, after the reduction phase the data were organised into categories based on the analysis; (3) interpretation using content analysis (Bardin 1991), creating patterns, concepts, and diagrams based on the objectives of the study. Categorical addition, categorisation, and coding were used to establish content categories. That is to say, after the establishment of categories, the interpretations were organised into a categorical sum based on the approaches of Bardin (1991), so we could later organise the interpretations from the categorical sum, to later be able to define patterns and schemes based on the objectives of the study; and (4) the ideological analysis, where the data were reflected upon and a discussion of the proposed theory was generated. This being the last phase after counting the categories, concepts and patterns from the content analysis, approaches were proposed in light of the theory, objectives and data that emerged from the study.

We analysed professors' secondary narratives and discourses from poststructuralist and feminist perspectives (Beasley 1999). This approach treated pluralities and differences, focusing on the subversion of categories such as identity, binaries, and relations, among others.

## 4. Analysis of Results

All professors interviewed agreed that schools impart norms, behaviours, and social patterns related to gender, class, and race aspects. Fred stated that, according to curricular research, images of social invisibility are transmitted from these spaces, contrasting with the visibility given to other characters: 'It [school] still transmits images of visibility, invisibility, thus; it still builds, it feels like we are still building a nation, the nation-state . . . '.

Stella mentioned that, regarding the curriculum,

> What is still being privileged is knowledge. To know content that is very connected to political history, economical history, and that, of course, sometimes references social history, but it is always very connected to political history . . . that sort of explain the difficulties that teachers have when one tells them *let's teach something else*, *let's give visibility to women and children*.

This transmission, they agreed, acts on different levels. From the critical paradigm, Giroux (2018) and Apple (1990) explained that the transmission of social structures of power privileges some contents over others. Teacher educators recognised that education, particularly history and social science teaching, is based on the production and reproduction of national feelings, on reproducing a sense of belonging to a certain territory with broad nationalistic sentiments.

Stella stated that curricular construction privileges the teaching of factual content over processes, problems, and the diverse versions that exist concerning historical events. Actually, the contents of history and social science have connections with political history, economical history, and, in some cases, social history 'from masculine, patriarchal, and white perspectives.' (Stella).

Thus, instead of transforming structures, participants agreed that history and social sciences teaching have produced and reproduced gender hegemonies, placing white and powerful men as protagonists of accounts and narratives. Following Foucault (1976), the school would be in charge of the standardisation of gender identities and expressions. That is why the social patterns that portray women as subordinate continue to be reproduced. Teachers' practices, their actions, discourses, and teaching strategies have the possibility of producing, reproducing, as well as generating changes to unequal structures.

For the professors interviewed, tradition is an obstacle for changing the approach in history and social science teaching programmes. As Vince stated, the dominant discourse standardises hegemonic and patriarchal structures of exclusion and subordination of women. This is how inequality, the processes, and the actions that have marginalised women are produced. As Vince added:

> Even if there is a discourse that talks about diversity, finally what happens in the classroom depends on the teacher . . . We talk about these men and these women, but we neither talk about all men nor about all women. Well, social history has kind of tried to incorporate this, but we talk about proletarian men and the rest is left aside . . .

Concerning this, tradition becomes a hegemonic construction with scarce variability. Conceptions of tradition have been utilised to thwart the inclusion of women, promoting the reproduction of historical and social gender structures (Villarroya et al. 2003). The protagonists are always men that have been prominent because of political, economic, and/or military power. Women who had any kind of power, or those women who have contributed to the construction of society with their work in everyday life, are invisible in these accounts.

Further, participants agreed when they stated that, despite the existence of a social tendency that promotes inclusion, most times the discourse, narratives, problems, and actions neither question nor reflect on the ways women are included, through the structures as well as the discursive accounts under which they are made visible. This discourse is built from patriarchal hegemony, and, therefore, excludes and subordinates not only women, but everything related to diversity, and that could generate changes in the androcentric traditional account. This way, inequality, stereotypes, and prejudices that revolve around participation and actions that have taken place from femininity are reproduced.

In addition to this, professors agreed that in accordance to what was aforementioned, immersed in an educational framework where patriarchal tradition predominates, when women are included, they become visible in actions that are 'proper to their gender' (Marolla 2019a, p. 23). That is, they are present in domestic chores, in the private sector and family care, amongst other classical stereotypes and prejudices that have surrounded women, in contrast to men, who are generally associated with war, politics, intellectuality and the public sector where apparently 'there were no women' (Marolla 2019b).

Foucault (1976) stated that tradition is the construct under which social structures and gender hierarchies have been validated and normalised; at the same time, these have generated oppression and social subordination over diversity. This comes into evidence, for example, in the construction of curriculums that reproduce dominant ideologies (Apple 1990; Giroux 2018; Valdés and Olavarría 1997). The exclusion of women is explained because the hegemonic culture is controlled by those who hold political, social, and economic power, which are white western men. They position their history, tales, and experiences as worthy of being historically transmitted in a way that is detrimental to women's narratives and memories.

For example, for Vince, the exclusion of diversity is directly related to the power of masculinity. Masculinity denies the opportunity to change the approaches and structures that can collaborate in the formation of multiple identities. According to participants, when a narrative or part of the history that was made by the actions of the traditionally excluded is revealed, it promotes identification with the purpose of generating empowerment and active political participation. For instance, when male and female students meet characters who have carried out different actions throughout history, an agency process leads them to become active citizens in their societies. From a gender perspective, education for citizen participation implies presenting the problem of inequalities and, at the same time, investigating the spaces and possibilities of fighting against inequalities caused by the same hierarchies and patriarchal social systems.

Regarding the above, Scott (1996, 2009) affirmed that history has been hierarchically constructed, positioning men as protagonists of history to the detriment of women's actions,

memories, and narratives (Levstik and Barton 2005; Sant and Pagès 2011). Therefore, participants were in complete agreement regarding the potential of social science teaching to change practices and contribute to the construction of knowledge applicable to teaching practices.

Vince delved into the importance of transforming traditional teaching practices; since history has been taught from an androcentric perspective, this has excluded not only women, but also homosexuals, lesbians, and transgender people, among other absent identities. The possibility of changing approaches is denied because it fosters the formation of multiple identities and therefore empowers learners to be an active and critical part of society. Vince stated:

> Because it homogenizes identities . . . because one can identify with history, because one recognises someone, and when one says that one wants to be like this someone, when I recognise a revolutionary person, so; if there are no women, you are denying the chance to build women's inherent historical memory, or, for example, when no gays or lesbians are part of history . . .

Therefore, the teaching of history and the social sciences must have an approach that privileges the treatment of problems, criticism, and the reflection on social inequalities. Especially focused on problems of gender discrimination. Valencia (2006) and González (2002) agreed that the structure and discourses that are transmitted in school reflect a passive and subordinate role for women compared to men, in which men have the leading role, as in stories from the teaching of history and the social sciences. It is fundamental, said Vince, that the constructions of what it is to 'be feminine' or 'to be masculine', and the power relations that have prevailed concerning these constructs are problematised. For Scott (1996), history and teacher education, built from male and endocentric conceptions, has reproduced a model of what it is to 'be a woman', and for that reason, schools have been producing and reproducing traditional roles based on gender categories.

Finally, professors affirmed that the ways in which the teaching of history and the social sciences are conceptualised are themselves obstacles, that is, the traditional frameworks that make up teaching. The official programmes on the teaching of history, in effect, are framed in the traditional perspectives that recognise masculinity and the dominant patriarchy. There is consensus that history has been socially constructed and, therefore, must be taught based on the actions and perspectives of the 'great men'. When working with women and diversity, the social representations of male and female students are broken by considering that what students are being taught is 'not history', since it is supposed to belong to an 'anecdotal' part of the 'really important' events, in which the actions of powerful men are highlighted.

In this regard, Tammy declared that there should be 'bridges' between the history that is taught and worked, and what is done from didactics. In particular, there should be links between academic knowledge and gender movements, focused on school learning and student training. Tammy acknowledged, as previously mentioned, that although her programme seeks to problematise women, including these approaches is still a pending task. Stella added that male and female teachers of didactics should ask themselves 'if history is not useful for solving the problems I have now, why should I teach it?'

Professors explicitly mentioned their intention to encourage understanding of the present and its diversity; however, classroom practices are contained within the scope of skills focused on the biographies of powerful men, wartime conflict, and male involvement in business activities. Although they position education for citizenship as one of the central objectives of their class, they do not operate with the lenses of citizen participation based on gender issues and the absence of women as historical subjects. Therefore, educational dynamics reflect the social constructions that have made women invisible and reproduced hierarchies, gender inequalities, and relegated women to the role of spectators. Professors spoke from a context in which affirming gender issues is not foregrounding the teaching of history and the social sciences. All this confirmed the approaches presented in the theoretical framework. Therefore, we argue it is essential that women and their history,

as well as gender issues, have a leading position as content and practice in the teaching of history and social sciences.

*The Chances to Include Women and Their History in Social Sciences Teaching*

Participants also highlighted the possibilities of including and working with women and diversity in general. This is beneficial in the context of constant problems and demands that educators must address against gender violence and in favour of social justice. The professors, therefore, agreed that the models used to build history must undergo constant reflection, and that the stories and narratives about women must become visible. They agreed that the teaching of history and social sciences must be rethought in order to include problems from a gender perspective, such as the relevant social problems that have caused, produced and reproduced discrimination, absence, and marginalisation of gender and women from historical categories. As Villuendas and López (2003) and Wittig (1992) said, questioning the discourses about women must account for the spaces of normalisation and naturalisation that have been transmitted regarding the roles and identities about 'being a woman' (Butler 1997; Schmeichel 2011; Schmidt 2012).

Crocco (2008) agreed with the comments made by participants that when women become invisible in history or, in some cases, are presented as an appendix to traditional male history, boys and girls do not have models to identify with and empower themselves; therefore, they do not fight against inequalities since they understand that their stories, women's stories, and everyday stories are not relevant. Their actions in the construction of history do not stand out, and they seem irrelevant to society. Stella, for example, explained that stories and narratives are predominantly masculine, but they do not represent all men. Rather, they are limited to those men who are white and who have had political, economic, and/or military power. When women are included, it is because they showed some kind of similarity with these men (Valdés and Olavarría 1997; Hess 2002; Badinter 1993; Banks 2008; González 2002; Gutiérrez 1998).

Stella argued that the characters included are 'the elite . . . a few men, presidents, soldiers who participated in some war, and who were important, who became heroes . . . from time to time a woman, but I think they are seen because they have masculine characteristics . . . The history of women is taught from a very anecdotal point of view' (Silva 2016; Heimberg 2005; Sánchez and Pintado 2006; Winslow 2013).

Polly, for example, commented that women are included in a very superficial way in history, as well as in didactics. The few women included are portrayed as 'great women', highlighting masculine characteristics rather than femininity itself (Crocco 2008; Fernández Valencia 2004; Sánchez and Pintado 2006). This generates a partial vision of the construction of history, provoking the conception that men, and specifically the patriarchy, have been the only participants in the construction of society and history (Scott 2009).

Fred emphasised the wide range of possibilities offered by the didactics of the social sciences to promote changes in traditional structures. The professor said that, from this area, it is possible to have a teacher education that encourages reflection and criticism of discourses, versions of history, and narratives: 'We have to make a change on historio-graphic discourses and narratives . . . they have to give evidence of the incorporation of visibility . . . ' (Fred 2016). Butler (2004, 2011), postulated, in accord with what participants have said, that there are ample spaces for the transformation of traditional narratives that have marginalised gender diversity, and proposed to society the questioning of versions, the normalisation, and naturalisation of the subordinate position that has been given to women, as well as diversity and multiple expressions of identity (Beasley 1999; Cerecer and Pagès 2014; Massip et al. 2020).

In order to put the aforementioned into practice, teachers could establish a break with traditional structures of teaching and learning history and social sciences, emphasising stories and narratives that relate how gender has been constructed and, therefore, how it has generated oppression against diversity (Ortega-Sánchez and Pagès 2018). The inclusion of women in the teaching of history and social sciences, according to participants, should

not be taught through themes, anecdotal data, or appendices to an androcentric history, but as an inclusion that proposes criticism and reflections about the topic (Woyshner 2002; Vavrus 2009) and the traditional and sexist structures under which history and its teaching have been built. Agreeing with these perspectives, Vince said that he would promote reflection aimed at a fairer and more inclusive society, respectful of diversity and human rights. For Vince, it is not a good idea to use structures that already exist to include women in history, but rather to problematise these structures and show the anecdotal and subordinate position to which women and diversity have been relegated. Therefore, structures could be promoted that would denature the hegemonic constructions that have been built by the patriarchy (Crocco 2008; Gutiérrez 1998; Villarroya et al. 2003; Hess 2002).

Professors agreed that positioning aspects such as criticism, reflection, and the denaturing of traditional structures of history in teaching could make women visible and make diversity visible in their daily actions, contributing to society (Marolla 2019a; Massip et al. 2020). Therefore, boys and girls could identify with these role models and understand that they can be part of and build history themselves, as fundamental protagonists in the construction of processes and spaces of historical change towards social justice, and most importantly, builders of a fairer future (Ross 2012; Levstik and Groth 2002; Cerecer and Pagès 2014).

Vince affirmed that didactics must consider how gender has been constructed and how structures of oppression towards diversity have been established. However, he argued that such structures should be raised not from the perspective of professors, but from students' own reflections on social and gender structures (Banks 2008; McIntosh 1983). Vince affirmed that the objective in his classes is to develop positions and critical thinking about gender oppressions, such as discrimination, prejudice, and stereotypes that perpetuate violence. The training should focus on: 'a teacher who can show the diverse, the reality, since in that logic there is a discourse that includes the theme of women, of diversity . . . there is also a diversity of identities . . . it seems that they are still seeing only a part of the diversity . . . ' (Vince 2016).

Participants agreed that a useful tool to achieve what has been said above is positioning and transforming a curriculum based on objective content into a curriculum based on relevant social problems (Sant and Pagès 2011), which makes women visible as actors in the construction of new structures outside the patriarchy (Scott 1996). This implies that the chronologies, phases, processes, concepts, dynamics, stories, and narrations of the actions of male and female characters should be rethought and transformed. Women and diversity cannot be included in the frameworks created by patriarchy because implicit oppression and subordination will continue to exist (Hahn et al. 2007; Heimberg 2005; Hess 2002; Hubbard 2013).

They understood that reflecting on the discourse and processes under which women have been included in the teaching of history and social sciences could represent a contribution to the fight against discrimination and gender inequality (Massip and Castellví 2019). The teachers believed that to fight the aforementioned problems, it is essential to reflect on the traditions that have led to such practices. For this reason, teacher educators expressed the problem of having a traditional history discipline based on great characters. They manifested the interest to include transformative perspectives on the teaching and learning of history from spaces that dismantle traditional structures of gender (Crocco 2008; Gràcia et al. 1993; Vavrus 2009).

The information provided by professors about their practices and training processes they carry out showed that society has established hierarchies where women occupy subordinate positions (Ortega-Sánchez and Pagès 2018; Marolla 2019a). History and social science teaching takes place under conditions that do not encourage reflection about problems that women have experienced and are currently experiencing. Participants emphasised the importance of creating spaces for the inclusion of women as a means to combat discrimination and inequality (Badinter 1993; Cerecer and Pagès 2014; Fernández Valencia 2004). On the other hand, teaching practices tend to base learning on curriculum

and traditional history, which generally represents the highly recurring presence of powerful men. Teaching practices in history and social sciences are based on political and/or economic issues, as well as on war, excluding women and their experiences, all of which cause inequalities to occur and to be reproduced from school spaces to social spaces and structures (Crocco 2015a; Luque and Rodríguez 2015; Lomas 2002; Heimberg 2005).

## 5. Discussion

Participants identified various advantages and limitations. Among the limitations, they agreed on the small and scarce role of the initial formative stages. They also mentioned the difficulties and the control imposed by the national curriculum and central administration, pushing them to teach the 'official' version of the story, seeking better results on standardised tests. Hence, the official curriculum is reaffirmed by participants as programmes do not provide spaces or possibilities for teachers to generate innovation, content, and practices from gender perspectives (Hess 2002; Crocco 2015b; Crocco 2008).

The importance that participants assigned to the inclusion of women is related to the possibility of social transformation (Beasley 1999; González 2002; Gutiérrez 1998). By posing questions and reflecting on discourse, space can be created for social transformation against gender inequality. The idea and approach that affirm the need to deconstruct traditional structures are reaffirmed, not only from the content with which they work, but also from the professors' own practices (Marolla 2019b; Ortega-Sánchez et al. 2020). This entire structure is part of the educational androcentrism that has perpetuated discrimination, the invisibility of women, and the rest of non-traditional gender structures (Foucault 1976; Hahn et al. 2007).

Participants commented on various advantages and limitations regarding the inclusion of women and the teaching of their history. Among the former, and in agreement with what is presented in the theoretical framework, they affirmed that in schools teachers are largely conditioned by curricular impositions (Vázquez 2003; Sánchez and Pintado 2006). Standardised tests are based on official history, tradition, and a focus on male and white characters. Learning the official history is what leads students to obtain good scores in these tests, and teachers have to teach said content in their classes, reinforcing such protagonists and historical dynamics (Scott 2009; Sant and Pagès 2011; Pinochet 2015).

Participants affirmed that the inclusion of women in the teaching of history and social sciences can encourage the development of spaces that motivate citizen participation (Ortega-Sánchez and Pagès 2018; Ross 2012). They recognised that they must challenge the discourses that make women visible and invisible. They also proposed that the problem lies in the discourse about women, which makes girls reproduce and perpetuate traditional models. The problem is not inclusion, but the type of discourse that makes women visible (Marolla 2019a, 2019b; Massip et al. 2020).

Among the relevance of the results obtained from the participants, we can affirm that the way women are portrayed in textbooks, in the school curriculum or the content of the class used by teachers transmits an androcentric image or perception of the experiences of women, youth, and adolescents (Woyshner 2002). Girls will identify with these images and reproduce them, that is, they will continue to maintain hierarchies and gender hegemonies in which they are positioned as subordinate to men (Crocco 2008; Fernández Valencia 2004; Lomas 2002). Similarly, if historical facts and history in general keep being transmitted through empowered white male protagonists who have been successful and have excelled in all aspects of society, white young boys and adolescents will grow up believing they are the only ones capable of leading movements and bringing about change in society (Ross 2012).

Participants' accounts were aligned with the theoretical perspectives presented in this paper, and even more, they affirmed that innovation must be the base of the practices they carry out. It should be noted that the innovation of the didactics to which they referred is not framed in the innovations proposed by the constructivist currents (Giroux 2018), but

refers to the deconstruction of the androcentric structures under which teaching has been conceived and reproduced (Villuendas and López 2003; Valencia 2006).

Joint efforts must be made towards curricular reform, eliminating sexism and androcentrism in schools (Villarroya et al. 2003). Teaching practices should consider different gender perspectives and focus on the roles, identities, and experiences of men and women in history (Scott 2009; McIntosh 1983; Schmidt 2012). For this reason, innovation in didactics is essential so it includes different materials for teaching and learning social sciences and history, as well as materials built on different gender perspectives (Villarroya et al. 2003; Vázquez 2003; Winslow 2013). It can be highlighted that teachers are willing to generate changes focused on the development of critical thinking, the formation of citizenship, and the questioning of androcentric identities in history (Sant and Pagès 2011; Ortega-Sánchez et al. 2020).

Therefore, professors interviewed in this paper proposed to create climates of tolerance, respect, inclusion, and gender diversity in classes (Marolla 2019a; Cerecer and Pagès 2014; Hubbard 2013). Participants suggested that teaching practices should not be limited to the transmission of content, but rather, to transform educational spaces to include differences and allow participation. Teachers argued that is important to empower students to manifest and fight against gender inequality (Gutiérrez 1998; Hahn et al. 2007). However, if they do not have the support of their respective schools and teachers, students will not be able to implement these transformations, since they will not have the space to do so (Woyshner and Schocker 2015; Schmeichel 2011; Cerecer and Pagès 2014; Levstik and Groth 2002).

From the educational exercise itself, it is necessary to promote teaching practices that promote social justice and the recognition of gender diversity (Ross 2012; Schmeichel 2011; Ortega-Sánchez 2017). All were willing to make commitments, generate changes and innovate in what they do to transform learning processes and society in general; however, if no updating tools are delivered from the curriculum or from teacher professional development, teaching will continue to reproduce inequalities (Villarroya et al. 2003; Sant and Pagès 2011; Marolla 2015, 2019a, 2019b).

Considering the advantages of the inclusion of women both in the curriculum and in the practices carried out by teachers, participants agreed that to clearly define and implement inclusive scenarios, teachers' willingness to contribute to this kind of formative process, the participation of educational centres and teaching practices are of vital importance (Winslow 2013; Villarroya et al. 2003; Vázquez 2003; Vavrus 2009). It is noteworthy that the professors thought that teacher education and teaching practices should be rethought to take into account women's experiences from a critical perspective, aimed at transforming gender inequality.

It is crucial that the organisation of the school is questioned from a perspective that provides space and favours the presence and participation of all, regardless of gender (Crocco 2015a). Although this is a complex goal, a perspective that fosters critical thinking about structural inequalities from a gender perspective should be central. One of the ideas that was raised is the creation of materials that address the experiences of women and propose reflection and consideration of current social constructions (Woyshner and Schocker 2015; Woyshner 2002; Hahn et al. 2007).

Teaching practices that incorporate objectives of social justice, inclusion, and citizen participation should be promoted, considering everyone as equals in their differences (Massip and Castellví 2019; Massip et al. 2020). The focus should be on citizen participation and commitment to the fight against gender inequalities, helping students to identify with social models and a history built around women, recognising and celebrating their actions and experiences (Valencia 2006; Ross 2012; Luque and Rodríguez 2015).

All professors interviewed recognised that one of the main limitations when trying to include women and their history in their teaching practices is the curriculum and the ministerial policies that normalise the discourse and strengthen the construction of identity and passive, subordinate, and non-participatory roles (Winslow 2013). Such roles,

in general, are associated to the construction of 'being a woman', as a role and gender subordinated by the predominance of men (Badinter 1993; Valdés and Olavarría 1997).

Traditional perspectives in teacher education and teaching practices do not deepen critical thinking and do not question the discourse transmitted through the teaching and learning of history. History teaching is focused on school curricular guidelines based on the transmission of androcentric structures of history (Marolla 2015) in which white, powerful and straight men are the protagonists of the stories. The pressure to teach these guidelines increases due to the pressures of the standardised tests, which are built from the ministerial requirements (Butler 2004; Fernández Valencia 2004; González 2002). It is not less important to highlight that in the Chilean context the results of standardised tests determine the financing schools receive from the government. The better the scores the greater are the resources received by the school, so the pressures to continue teaching traditional perspectives are rooted in training practices (Marolla 2019a, 2019b; Ávalos 2014).

When women are included in ministry policies and within the curriculum and traditional structures of history and the social sciences, they appear as 'spectators' (Woyshner and Schocker 2015; Woyshner 2002) of history. That is, in a way annexed to a story starring powerful men (Beasley 1999). Due to social hierarchies, produced and reproduced both by society and by educational structures, the absence of women and sexual dissent has been normalised in favour of the construction of history centred on male protagonists. In this context, education is a transmission of the social order itself that is reproduced in society (Bourdieu and Passeron 2018; Heimberg 2005).

Therefore, when women are included in practices and the curriculum they are reflected similarly to the roles traditionally assigned to men (Gràcia et al. 1993; Gutiérrez 1998; Hess 2002; Hubbard 2013). In other words, they include women who have also had political, economic, and warfare power, generating assimilation and reproduction of the power structures and normalised hierarchies that maintain the status quo concerning gender perspectives (Sant and Pagès 2011; Pagès and Villalón 2013; Massip et al. 2020).

The current economic system collaborates to normalise class and gender inequality, linking women to private and domestic roles. This is contrary to the emphasis given to men as leaders both in public spaces and in the workplace. This is evidenced in current investigations (Banks 2008; Hahn et al. 2007; Crocco 2008; Sánchez and Pintado 2006). Women and their stories, when they become visible in educational processes, are portrayed in public spaces, assuming roles traditionally assigned to women (Beasley 1999; Gutiérrez 1998; Gràcia et al. 1993).

## 6. Conclusions

As participants have stated, it is essential to rethink how teachers are trained, including spaces for reflection and analysis on the discourse under which women and the range of gender diversity have become invisible and excluded. Teacher education must consider the invisibility of women and other groups and suggest new ways of classifying history, questioning current social constructions and bringing new problems to the classroom.

As the different participants in the current study agreed, teaching practices should focus on aligning discourse with action. Participants suggested a framework of possibilities that should be translated into practices aimed at transforming gender inequality. One way to do this is to reflect on existing structures and reveal the ultimate purpose and objectives sought, for example, by the curriculum, textbooks, and school administration (González 2002; Gutiérrez 1998; Levstik and Barton 2005; Lomas 2002; Villarroya et al. 2003).

Teacher education should provide ideas and tools to promote teaching practices that focus on aspects of emancipation and the visibility of non-traditional roles for women. The formative processes of teachers must analyse and reflect on the structures on which history has been built. If they could promote more reflective teaching practices that encompass the teaching and learning of history and the social sciences, they would have as much potential to deconstruct social constructs as they now have to promote the reproduction of inequality. This is essential to revealing the implicit meaning in discourses and the structures of gender,

class, or ethnicity included in the teaching of the aforementioned subjects. Teachers must be critical of their own actions and be able to understand the perspectives under which certain processes, discourses, or protagonists become visible or invisible.

The deconstruction of social structures in teacher education must enable teachers to rethink the final objectives of teaching practices and learning and deconstruct gender hierarchies and the subordination that has led to the marginalisation, abuse, and obscuring of women. Performativity (Butler 1997, 2011, 2016) can constitute a theoretical and practical tool used to collaborate in uncovering how gender structures have worked and how new alternatives can be built to combat monopolised power in the form of patriarchy.

Formative processes should consider the use of materials and alternative theoretical perspectives to those traditionally used. Materials that gather and question content from a gender perspective can provide options in the struggle against oppression and the reproduction of roles and identities passed on, worked on, and currently existing in classrooms.

Questioning current gender construction and structures is key to promoting transformation of the status quo. Questioning them should create spaces with a climate of real possibility in the classroom. These spaces should focus on the participation and empowerment of students in the struggle against gender inequality.

*Limitations*

Regarding limitations, it is possible to mention the scale covered by the study. That is, it is only analysed and discussed on the basis of a specific sample of Chilean professors. Hence, there is no possibility of generalising the findings, as well as of transferring such ideas to other contexts and realities. Although the contexts and problems addressed in the study for gender reasons are similar, the results discussed are specific to the case study carried out. We argue, though, that the study presents the possibility to apply some paths, ideas, tools and spaces of possibility that can be applied in other educational contexts.

In addition, among the limitations, it is possible to point out the large amount of information provided by the interviews. It is difficult to select from and analyse the large amount of data that was available. Despite the existence of different sources of educational material stemming from the social sciences that considered gender perspectives in teaching practices and learning, the experiences and problems of women are not included in formative processes or teaching practices.

Finally, it is recommended that studies that propose the analysis of gender perspectives in the teaching and learning of history and social sciences delve into the practices carried out by teachers.

It can be mentioned that the main projection of the research is related to two perspectives. The first is being able to replicate the methodological structure in new studies that delve into other contexts and realities from the purposes established by the study. Second, it is necessary to carry out quantitative studies on a larger scale on the reality and conceptions of teachers regarding the inclusion of women and gender perspectives in the teaching of history and social sciences.

In the same way, it is relevant that, from the ideas raised in the study, studies are proposed that investigate the teaching practices carried out by the teaching staff from the inclusion of gender perspectives. On the other hand, it is argued that the objective of the studies on such themes has an impact on the teacher practices. Likewise, the changes proposed in the paper are related to deep-rooted structures of society.

Participants recognised the questions being asked and the gender structures they themselves are part of. They recognised themselves as part of the problem; however, they put forward a scenario that mobilises the possibility of transforming inequality. These professors were willing to transform inequality and work collaboratively and inclusively with everyone participating and forming part of teaching and learning processes.

However, it can help the teachers to reflect on their own thinking about gender structures, and from such perspectives, to promote changes in the practices that have

contributed to the reproduction of inequalities. In the same way, the findings found in the study can help those in charge of the construction of teacher training programmes and the history and social sciences curriculum developers to reflect on the perspectives and structures that are being made visible. That is, the spaces for change and transformation that arise from the programmes and the curriculum, as well as the spaces that continue to reproduce the structures of inequality. In this way, as stated before, it is optimistic to expect profound changes in the face of structures, hierarchies, and gender hegemonies that have traditionally been perpetuated throughout history; however, studies such as this one represent spaces of possibility and transformation both for teachers as well as for management teams.

### 7. Ethics Statement

This study was carried out in accordance with the recommendations of Research Ethics in Education of the University of the Americas, Chile. The protocol was approved by University of the Americas, Chile. All subjects gave informed consent in accordance with the Declaration of Helsinki.

**Author Contributions:** Conceptualization, J.M.G. and J.C.M.; methodology, J.M.G.; J.C.M. and R.M.d.S.; software, J.C.M.; validation, J.M.G. and R.M.d.S.; formal analysis, J.M.G., J.C.M. and R.M.d.S.; investigation, J.M.G., J.C.M. and R.M.d.S.; resources, J.M.G.; data curation, J.C.M.; writing—original draft preparation, J.M.G., J.C.M. and R.M.d.S.; writing—review and editing, J.M.G., J.C.M. and R.M.d.S.; visualization, R.M.d.S.; supervision, J.C.M.; project administration, J.M.G. All authors have read and agreed to the published version of the manuscript.

**Funding:** This research was funded by University of Americas, Chile, grant number UA1016. The main researcher is Professor Jesús Marolla Gajardo (University of Americas, Chile).

**Institutional Review Board Statement:** The study was conducted according to the guidelines of the Declaration of Helsinki, and approved by the Ethics Committee of University of the Americas (REX Nº123120 08/06/2020).

**Informed Consent Statement:** Informed consent was obtained from all subjects involved in the study.

**Data Availability Statement:** Not applicable.

**Conflicts of Interest:** The authors declare no conflict of interest.

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
