# Peer review of "Chilean Teacher Educators’ Conceptions on the Absence of Women and Their History in Teacher Training Programmes. A Collective Case Study"

_socsci, doi:10.3390/socsci10030106_

Round 1

Reviewer 1 Report

Thank you for submitting your paper for consideration. This is an interesting study that has the potential to make a valuable scientific contribution. However, several shortcomings and recommendations should be addressed, as the current version manuscript is not suitable for publication due to several reasons.

1) First of all, the manuscript requires extensive language editing by a native English speaker (e.g., starting from the Abstract itself, there are several ambiguous words, unclear phrases, and misuse of expressions; ”In special,”; ”From different theoretical perspectives in education it has been manifested the urgency (...)”; "Social studies, as an educational area, is concerned with this problematic"; "Some studies such as (...) agree that women and their history have been worked from the margins and absences, that is, many times as appendices or curious facts in a history built from masculinity."; "Such history and teaching of history, the authors agree in this way, highlights the actions of powerful men in politics, the economy and war."; "There existed efforts to represent"...etc)

Lines 38-42: excessively long and unclear

Lines 43-45: References needed

Lines 23-27: unclear and unstable choices of references list;

Many paragraphs in the Introduction are repetitive, describing more than once patriarchal and androcentric structures and their implications. Therefore, I advise the authors to entirely revise this section.

2. The "2.3. Gender and education in Chile" section is unclear and I advise the authors to reconsider its usage.

3. There are no clear details related to specific assumption & participants demographics.

Finally, the Discussion and Conclusion part comes before the MEthods and Procedure section. It's hard to follow and understand the entire study. 

Author Response

Point 1: Thank you for submitting your paper for consideration. This is an interesting study that has the potential to make a valuable scientific contribution. However, several shortcomings and recommendations should be addressed, as the current version manuscript is not suitable for publication due to several reasons.

Response 1: Thank you very much for your revision. The language editing issues have been corrected and we appreciate your comments.

Point 2: First of all, the manuscript requires extensive language editing by a native English speaker (e.g., starting from the Abstract itself, there are several ambiguous words, unclear phrases, and misuse of expressions; ”In special,”; ”From different theoretical perspectives in education it has been manifested the urgency (...)”; "Social studies, as an educational area, is concerned with this problematic"; "Some studies such as (...) agree that women and their history have been worked from the margins and absences, that is, many times as appendices or curious facts in a history built from masculinity."; "Such history and teaching of history, the authors agree in this way, highlights the actions of powerful men in politics, the economy and war."; "There existed efforts to represent"...etc)

Lines 38-42: excessively long and unclear

Response 2: Lines 38-42 were corrected by changing the previously discussed ideas and including new perspectives.

Point 3: Lines 43-45: References needed

Response 3: The literature was reviewed and necessary references about the ideas commented were included.

Point 4: Lines 23-27: unclear and unstable choices of references list

Response 4: The reference list was modified adjusting it to the document content only.

Point 5: Many paragraphs in the Introduction are repetitive, describing more than once patriarchal and androcentric structures and their implications. Therefore, I advise the authors to entirely revise this section.

Response 5: The introduction was revised removing redundant ideas. The content was adjusted to the fundamental aspect of each idea that was meant to get across.

Point 6: 2. The "2.3. Gender and education in Chile" section is unclear and I advise the authors to reconsider its usage.

Response 6: Section 2.3 was deleted. A few of your ideas were included in other sections of the document. However, it was considered that a few of the ideas provided were focused on another topic that is why only the ones that followed the main topic of the document were prioritised.

Point 7:  There are no clear details related to specific assumption & participants demographics.

Response 7: With the purpose of being clearer, participants’ information Table 1 was added including specific information about the teachers.

Point 8: Finally, the Discussion and Conclusion part comes before the MEthods and Procedure section. It's hard to follow and understand the entire study.

Response 8: The methodology and process section was added after the theoretical framework section. In addition to these changes the name of the section was changed to ‘Methodology’.

Reviewer 2 Report

First of all, many thanks to the authors for the opportunity to read their work.
The topic addressed is an emerging one.
The theoretical framework goes too deeply into the social background of the topic in question. 
However, it does not go very deeply into issues related to education, even though it is the central theme of the paper. 
Therefore, it is recommended that the background of the topic in the field of education be explored in more depth in order to make the theoretical framework coherent with the research.
Gender and education in Chile offers a personal assessment of the Chilean education system.  But it would be necessary to substantiate this issue with authors and theorists in order to reinforce these ideas.
On the other hand, in terms of the Chilean educational context, special mention is made of the organisation of the system divided by gender as one of the issues that sustains the situation and segregation of women in educational and social aspects. However, we consider that in order to address the educational issue, it would be interesting to broaden this vision to include teaching practices, curricular approach, etc.  So we consider that this issue should be expanded with previous research. 
Also, the research topic is approached from the field of history and social sciences and the theoretical framework does not deal with this perspective in its background. It would be desirable to provide more international references on the topic, which would broaden the perspective on the subject.
The results section contains too much information, therefore, it would be advisable to clarify or summarise the content of the results with an outline or conceptual map that offers the information in an easier and more attractive way.
Materials and Methods should be placed before the discussion and conclusions section in accordance with the traditional structure and organisation of a scientific article.
The discussion section is very well argued and constructed. It is therefore necessary to reduce its length and expressly deal with the conclusions obtained from the topic after the research and the response given to the objectives, as well as the limitations of the study, future lines of research and the transfer of the results, i.e., the usefulness or practical application of the findings.

Author Response

Point 1: First of all, many thanks to the authors for the opportunity to read their work.

The topic addressed is an emerging one.

The theoretical framework goes too deeply into the social background of the topic in question.

However, it does not go very deeply into issues related to education, even though it is the central theme of the paper.

Therefore, it is recommended that the background of the topic in the field of education be explored in more depth in order to make the theoretical framework coherent with the research

Response 1: We made changes to the theoretical framework, however; it is fundamental to thoroughly explain the social background since gender issues are closely related to society’s structures and how these are reproduced in educational practices. This is

how society and education are connected. Concerning educational research on this topic, unfortunately there is no specific research about History and Social sciences teaching from a gender perspective.

Point 2: Gender and education in Chile offers a personal assessment of the Chilean education system.  But it would be necessary to substantiate this issue with authors and theorists in order to reinforce these ideas.

On the other hand, in terms of the Chilean educational context, special mention is made of the organization of the system divided by gender as one of the issues that sustains the situation and segregation of women in educational and social aspects. However, we consider that in order to address the educational issue, it would be interesting to broaden this vision to include teaching practices, curricular approach, etc.  So we consider that this issue should be expanded with previous research.

Also, the research topic is approached from the field of history and social sciences and the theoretical framework does not deal with this perspective in its background. It would be desirable to provide more international references on the topic, which would broaden the perspective on the subject.

Response 2: Due to the argumentative line and because it did not directly relate to the documents’ ideas, the section ´gender and education in Chile’ was removed, and a few of the ideas from this section were included in other sections.

We agree that it is necessary to deeply investigate and study the Chilean educational system. Therefore, it has been decided to remove the ideas that are not related directly with this study. This, on one hand, as a way to safeguard the argumentative integrity, and on the other hand, we recognize that what was stated proposed other topics.

According to the study’s context a well as the participants’, research from different Western scenarios has been selected in Spanish as well as in English. This in order to add theoretical diversity to the study.

Point 3: The results section contains too much information, therefore, it would be advisable to clarify or summarize the content of the results with an outline or conceptual map that offers the information in an easier and more attractive way.

Response 3: A few ideas in the results section were summarized. However, no diagram was added but ideas were put in order and topics were prioritized.

Point 4: Materials and Methods should be placed before the discussion and conclusions section in accordance with the traditional structure and organization of a scientific article.

The discussion section is very well argued and constructed. It is therefore necessary to reduce its length and expressly deal with the conclusions obtained from the topic after the research and the response given to the objectives, as well as the limitations of the study, future lines of research and the transfer of the results, i.e., the usefulness or practical application of the findings.

Response 4: The order of the sections was changed, the position of materials before of the theoretical framework. The discussion and conclusions sections were divided leaving them as separate sections. In these sections ideas were taken out and added with the objective to give depth to what is discussed. Specifically, future lines of investigation and practical usefulness of the findings were added to the conclusion section.

Reviewer 3 Report

Dear Authors:
The manuscript explores an interesting and relevant topic, if "The world is changing – education must also change."
The theoretical framework is scientifically robust. However, an effort should be made to reinforce the existence of bibliographical references from the last three years. In your manuscript, this represents only 5%. 
We note that there are some aspects that need improvement.
You should correct the sentence in lines 11-12: “The methodology utilized is Collective Case Studies. The methodology is positioned (…)”
The structure of the manuscript is not correct. 
The "Introduction" should clarify the interest of the study that was developed.
Section 3 should be "Materials and Methods". The content presented for "Materials and Methods" does not fully correspond to what we read in the "Results" section. Otherwise, for example, revisit what you wrote in lines 582-585, 628-640 and 641-643. What you wrote in lines 645-650 should be in the "Introduction" section. The "Results" section should be section 4. By the way, it would be more correct to name this section "analysis of Results" and match it with the methodology (categories of analysis).
It seems to me that your subsection 3.1 is within the scope of the "Discussion" section.
It would be advisable to create a final section with only the main conclusions of the study, separating in another section the "recommendations and limitations". My suggestion is to create the "Discussion" section (section 5).

Sincerely,
Rev

Author Response

Point 1: The manuscript explores an interesting and relevant topic, if "The world is changing – education must also change."

The theoretical framework is scientifically robust. However, an effort should be made to reinforce the existence of bibliographical references from the last three years. In your manuscript, this represents only 5%.

Response 1: Thank you very much for your comments and suggestions that without a doubt have helped improve the document we submitted.

New investigations have been included in the mentioned references included in the theoretic framework. Due to the few studies on the subject, better studies have not been found. This is the reason why this study is of great contribution to this under studied area.

Point 2: We note that there are some aspects that need improvement.

You should correct the sentence in lines 11-12: “The methodology utilized is Collective Case Studies. The methodology is positioned (…)”

Response 2: The mentioned lines that needed improvement, such as the organization of the document, were restructured.

Point 3: The structure of the manuscript is not correct.

The "Introduction" should clarify the interest of the study that was developed.

Section 3 should be "Materials and Methods". The content presented for "Materials and Methods" does not fully correspond to what we read in the "Results" section. Otherwise, for example, revisit what you wrote in lines 582-585, 628-640 and 641-643. What you wrote in lines 645-650 should be in the "Introduction" section. The "Results" section should be section 4. By the way, it would be more correct to name this section "analysis of Results" and match it with the methodology (categories of analysis).

Response 3: The introduction section has been improved by clarifying the interest of the study that we conducted.

Suggestions to lines 582-585, 628- 640 and 645- 650 have been taken into consideration by modifying ideas and including others in the introduction as it was suggested.

Point 4: It seems to me that your subsection 3.1 is within the scope of the "Discussion" section.

It would be advisable to create a final section with only the main conclusions of the study, separating in another section the "recommendations and limitations". My suggestion is to create the "Discussion" section (section 5).

Response 4: It has been taken into consideration the suggestion to include the section of ‘result analysis’.  The new section ‘discussion’ has been created and it is a separate section from the newly created ‘conclusions’ section. The section of ‘recommendations and limitations’ has been added to the conclusions following the suggestions from another reviewer.

Round 2

Reviewer 1 Report

Thank you for revising the paper. The authors addressed all my concerns, and I can now recommend the paper for publishing. 

Author Response

Dear reviewer

We greatly appreciate your suggestions, which have undoubtedly contributed to the improvement of the writing. All proposed enhancements have been included.

Reviewer 3 Report

Dear authors,

The manuscript is much improved.
However, there are still some aspects that need your attention.
Check the names of the sections (e.g., "44. Analysis of resultsresults") and the fictitious names of the participants (e.g. "StellaStella").
I still think that the objectives (lines 213-219) should appear in the "Introduction" section, i.e., in line 53.
Regarding the subsection "limitations", I note that recommendations and limitation are mixed up. In fact, the last paragraph of the "conclusion" section already refers to limitations of the study.

Rev

Author Response

Dear reviewer

We appreciate the comments made. They have undoubtedly contributed to the improvement of the writing.
Fixed repeated fictitious names issues.
the objectives have been moved to the introduction as suggested.
the section on conclusions and limitations has been modified, specifically clarifying that the document is framed in what the subsection deals with.
